# OPTIMIZED GATED DEEP LEARNING ARCHITECTURES FOR SENSOR FUSION

## ABSTRACT

Sensor fusion is a key technology that integrates various sensory inputs to allow for robust decision making in many applications such as autonomous driving and robot control. Deep neural networks have been adopted for sensor fusion in a body of recent studies. Among these, the so-called netgated architecture was proposed, which has demonstrated improved performances over the conventional convolutional neural networks (CNN). In this paper, we address several limitations of the baseline negated architecture by proposing two further optimized architectures: a coarser-grained gated architecture employing (feature) group-level fusion weights and a two-stage gated architectures leveraging both the group-level and feature-level fusion weights. Using driving mode prediction and human activity recognition datasets, we demonstrate the significant performance improvements brought by the proposed gated architectures and also their robustness in the presence of sensor noise and failures.

## 1 INTRODUCTION

Sensor fusion is an essential technology to autonomous systems such as self-driving cars and mobile robots. In advanced driver-assistance systems (ADAS), many sensors such as cameras, ultrasonic sensors, and LiDARs are utilized for enhanced safety and driving experience. Sensor fusion based vehicle and robot control technologies have been explored (Vargas-Meléndez et al., 2016; Jain et al., 2016; Garcia et al., 2017; Bohez et al., 2017; Patel et al., 2017). In addition, devices like smartphones and smartwatches typically integrate a number of sensors, making these devices a suitable platform for running sensor fusion applications such as activity recognition. Several sensor fusion techniques for activity recognition have been proposed (Yurtman & Barshan, 2017; Dehzangi et al., 2017; Zhao & Zhou, 2017; Gravina et al., 2017).

More specifically, in Bohez et al. (2017), a deep reinforcement learning based sensor fusion algorithm is discussed for robot control. A Kuka YouBot is used with multiple LiDAR sensors for simulations. In terms of sensor fusion, early fusion which concatenates all inputs as feature planes is compared with three other fusion techniques: concatenating convolution layer outputs, reducing the concatenated features with a 1x1 convolution, and accumulating convolution outputs. However, sensor noise and failures are not considered in this work. In addition, due to the fact that only the same type of sensory inputs is used in the experiments, the performance of sensor fusion based on different kinds of sensory inputs is unclear.

Among sensor fusion techniques employed in automobiles, Vargas-Meléndez et al. (2016) exploit neural networks with a Kalman filter for vehicle roll angle estimation, and show the advantage of using an inertial measurement unit (IMU) without additional suspension deflection sensors. Jain et al. (2016) consider a sensor-rich platform with a learning algorithm for maneuver prediction. Long short-term memory (LSTM) networks, which are a type of recurrent neural networks (RNN) are used with sensory inputs from cameras, GPS, and speedometers. Garcia et al. (2017) propose joint probabilistic data fusion for road environments. However, neither a multiplicity of sensory inputs nor sensory noise and failures are considered. The adopted architecture is simplistic where input data are only fused in one layer.

In the field of wearable devices, Yurtman & Barshan (2017) utilize early fusion, which concatenates sensory inputs. With this simple fusion approach, classical supervised learning methods such as Bayesian classifiers, k-nearest-neighbors, support vector machines, and artificial neural networks

are compared. Dehzangi et al. (2017) use deep convolutional neural networks with IMU data. Zhao & Zhou (2017) use a CNN with angle embedded gate dynamic images, which are pre-processed inputs for gait recognition. Gravina et al. (2017) summarize three fusion methods, namely, data, feature, and decision level fusion. However, effective sensor fusion network architectures for coping with sensor failures are not deeply investigated.

In terms of sensor fusion architectures, Patel et al. (2017) propose a so-called netgated architecture in which the information flow in a given convolutional neural network (CNN) is gated by fusion weights extracted from camera and LiDAR inputs. These fusion weights are used for computing a weighted sum of the sensory inputs. The weighted sum passes through fully connected layers to create a steering command. The gated networks (netgated) is shown to be robust to sensor failures, comparing to basic CNNs. However, a deep understanding of the relationships between sensory inputs, fusion weights, network architecture, and the resulting performances are not examined.

The main objective of this paper is to propose optimized gated architectures that can address three limitations of the baseline netgated architecture of Patel et al. (2017) and investigate how different fusion architectures operate under clean sensory data and in the presence of snesor noise and failures. Our main contributions are:

- Propose a new coarser-grained gated architecture which learns robustly a set of fusion weights at the (feature) group level;

- Further propose a two-stage gating architecture which exploits both the feature-level and group-level fusion weights, leading to further performance improvements.

- Analyze the characteristics of the proposed architectures and show how they may address the limitations of the negated architecture in terms of inconsistency of fusion weights, over-fitting, and lack of diverse fusion mechanisms.

By utilizing driving mode prediction and human activity recognition datasets, we demonstrate the significant performance improvements brought by the proposed architectures over the conventional CNN and netgated architectures under various settings including cases where random sensor noise and failures are presented. Empirical evidence is also analyzed to help shed light on the underlying causes that may be responsible for the observed improvements of the two proposed architectures.

## 2 THE BASELINE NETGATED ARCHITECTURE AND ITS LIMITATIONS

### 2.1 THE NETGATED ARCHITECTURE

The netgated architecture proposed in Patel et al. (2017) offers a promising approach for sensor fusion. This architecture was proposed under the context of unmanned ground vehicle(UGV) autonomous driving with convolutional neural networks with two sensors. A more general version of this architecture (with five sensors/features) is depicted in Fig. 1. In Patel et al. (2017), data from two sensory inputs, i.e. camera and LiDAR are processed individually through convolutional (conv) layers, pooling layers, and fully connected layers. The outputs from the fully connected (FC) layers ( e.g. "FC-f1" to "FC-f5" in the first dashed box in Fig. 1), are concatenated and then fused by another FC layer (e.g. "FC-con" in Fig. 1), where two feature-level fusion weights are created. Feature-level fusion weights are originally referred to as scalars in Patel et al. (2017). Note that each fusion weight is a scalar value and that in Fig. 1 five feature-level fusion weights are extracted which are the outputs of the "FC-con" layer. These fusion weights are multiplied with the corresponding outputs from the feature-level FC layers, i.e. the first dashed box, which is duplicated for better illustration of data flow in Fig. 1. Finally, these weighted feature outputs are fused by the last FC layer (i.e. "FC-out"), which produces the final prediction decision.

The negated architecture is interesting in the sense that the extracted feature-level weights may be conceptually thought as a "gated" variable for each feature input and that a sensory input (feature) may be shut off from the network by the zero-valued fusion weight when it is corrupted by noise or sensor failures. As such, this architectures may be promising in providing robust sensor fusion capability.

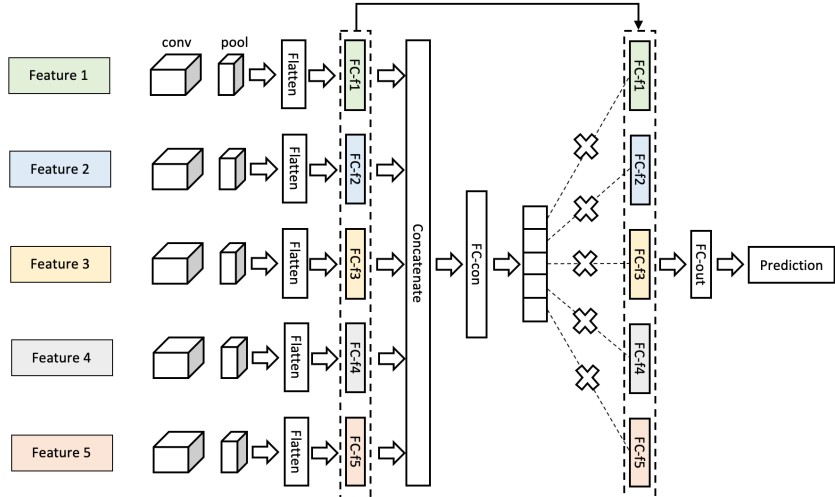

Figure 1: The Netgated CNN architecture proposed in Patel et al. (2017). Extended to include five features here.

## 2.2 Limitations of the Basic Netgated Architecture

The negated architecture offers as an appealing end-to-end deep learning solution. Nevertheless, partially due to its end-to-end black-box nature, this architecture has several limitations as discussed below.

**Inconsistency of Fusion Weights.** First, consider the situation in which there are $N$ input features $f_1, f_2, \cdots, f_N$ with the corresponding feature-level fusion weights $fw_1, fw_2, \cdots, fw_N$. As in Fig. 1, the feature-level fusion weights are produced by the "FC-con" layer based on fused information from all inputs. As a result, an extracted fusion weight might not fully correspond to the corresponding feature due to the information sharing among all features. As we have observed in our experimental studies, there exist cases where the feature with the largest extracted fusion weight does not represent the most critical feature for the learning task. While the ranking of the feature-level weights may reflect the relative importance of the corresponding features to a certain degree, the association between the two is not always consistent. In this paper, we refer to this phenomenon as *inconsistency of fusion weights*. It can be well expected that inconsistency of fusion weights may adversely affect the overall prediction accuracy, particularly when the fusion weights for certain noisy or corrupted features are not robustly learned by the network, resulting misleadingly large fusion weight values.

**Potential Over-fitting.** Furthermore, for applications where many features need to be fused, using the same number fusion weight values introduces many additional parameters that shall be learned properly in the training process, making over-fitting of the model easier to occur. This situation further exacerbates due to the potential occurrence of inconsistency of fusion weights.

**Lack of Additional Fusion Mechanisms.** Finally, in the architecture of Fig. 1, apart from the learning of fusion weights, fusion of raw input features is done in a simplistic manner, i.e. by the last fully connected layer "FC-out". Nevertheless, there exist more powerful raw input fusion mechanisms which could potentially lead to additional performance improvements.

We address the above limitations of the baseline netgated architectures by proposing two extensions: a coarser-grained architecture and a hierarchical two-stage architecture, referred to as the Feature-Group Gated Fusion Architecture (FG-GFA) and the Two-Stage Gated Fusion Architecture (2S-GFA), respectively, described in the following sections.

## 3 Feature-Group Gated Fusion Architecture (FG-GFA)

To address the aforementioned limitations of the baseline netgated architecture, we first explore the coarser-grained architecture, namely, Feature-Group Gated Fusion Architecture (FG-GFA) as in Fig. 2, where for illustration purpose two feature groups are shown with the first group having three features and the second group two features. In general, a given set of $N$ input features $f_1, f_2, \cdots, f_N$ may be partitioned into, say $M$, feature groups $FG_1, FG_2, \cdots, FG_M$. As one specific example of this architecture, all features in a feature group are concatenated first, and then passed onto a convolution layer and a pooling layer. After going through the corresponding FC layer ("FC-g1" or "FC-g2" in Fig. 2), the processed information from all groups are concatenated and then passed onto an FC layer ("FC-con" in Fig. 2) whose outputs are split into $M$ group-level fusion weights. The fused information of each group e.g. 'FC-g1' or "FC-g2" in Fig. 2, is multiplied by the corresponding group-level fusion weight ("FC-g1" or "FC-g2" are again duplicated in Fig. 2 to better illustrate the information flow). All weighted group-level information is combined and then processed by the final FC layer ("FC-out" in Fig. 2) to produce the prediction decision. The configuration and the number of layers at each key step of the FG-GFA may be chosen differently from the specific example of Fig. 2.

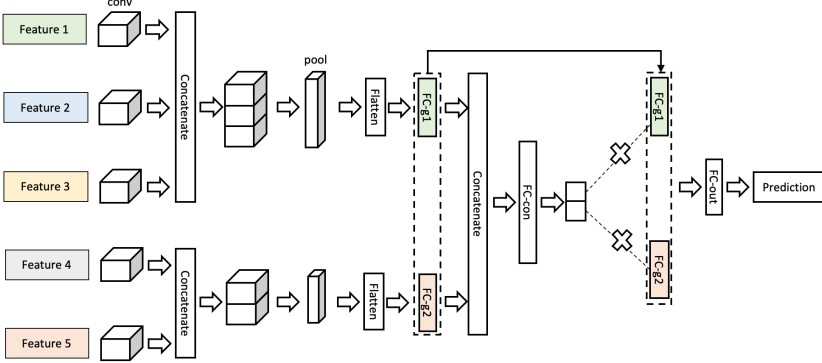

Figure 2: The proposed feature-group gated fusion architecture (FG-GFA).

We now comment on the key differences between the FG-GFA architecture and the baseline netgated architecture. First of all, in addition to the final fusion operation of the "FC-out" in Fig. 2, we have performed additional early fusion of sensory inputs within each group. The outputs of such within-group fusions are combined to produce a smaller number of group-level fusion weights. Furthermore, the extracted group-level weights are used to multiply the corresponding fused group feature information, not the individual feature information. These characteristics of FG-GFA introduce different types of fusion mechanisms into the network. Second, since now fusion weights are extracted only at the group-level, fewer weights need to be learned compared with the baseline architecture. The fact that there are now a less number of tuning parameters and that early fusion takes place within each group might reduce the likelihood of stucking the training process on local minima. As a result, it may be possible to mitigate the issues of inconsistency of fusion weights and potential over-fitting. As will be demonstrated in the experimental study, FG-GFA leads to significantly more robust learning of fusion weights at the group level, i.e. existence of noisy or corrupted features in a group can be much more reliably reflected in the corresponding reduction of the group-level fusion weight. As a result, group-level fusion weights become a more reliable indicator of the quality of the sensory inputs in each group. Accordingly, we have empirically observed improved performance brought by FG-GFA as demonstrated later.

## 4 The Proposed Two-Stage Gated Fusion Architecture (2S-GFA)

In this hierarchical fusion architecture, we combine the baseline netgated architecture that leans the feature-level fusion weights and the proposed feature-group gated fusion architecture (FG-GFA) that extracts group-level fusion weights into two stages. The 2S-GFA architecture is illustrated in Fig. 3 where the first three features are in group 1 and the remaining two features are in group 2.

The upper portion of the network extracts five feature-level fusion weights based on splitting the outputs of the FC layer "FC-con", in the same way as in the baseline netgated architecture. The smaller sub-network at the bottom of Fig. 3 reuses outputs from the first stage of conv layers on the top of the figure that pre-process all sensory inputs individually. Then it concatenates the pre-processed feature information within each group. It produces two group-level fusion weights by splitting the outputs of the FC layer "FC-con-g", shown by the red and yellow squares for the two groups, respectively. For each feature input, the product of its feature-level fusion weight and the group-level fusion weight defines its final feature weight, which is used to multiply the processed feature information, e.g. "FC-f1"" to "FC-f5" in Fig. 3. Then, all weighted feature information are fused together by the FC layer "FC-out" which produces the final decision.

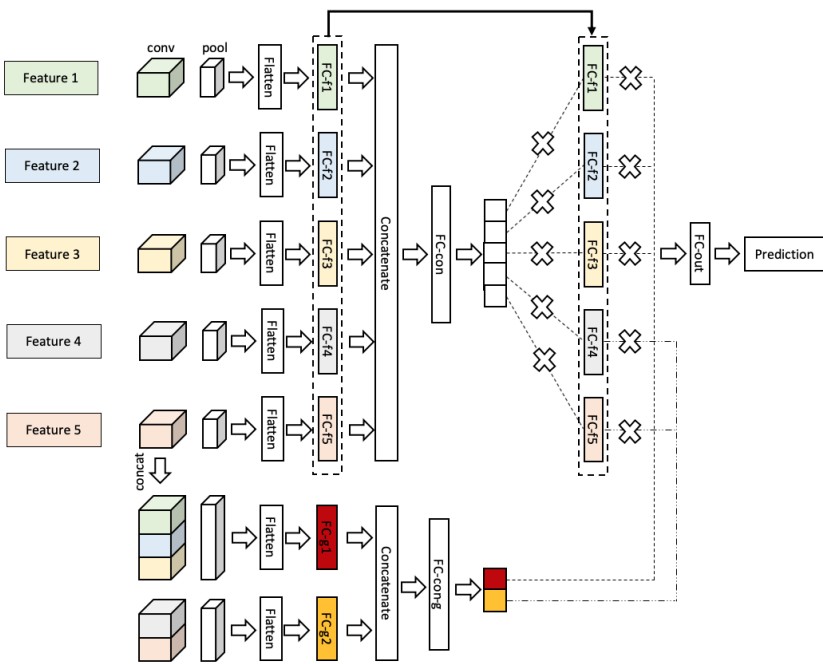

Figure 3: The proposed two-stage gated fusion architecture (2S-GFA).

Since 2S-GFA integrates the essential ingredients of the baseline netgated architecture and the group-based FG-GFA architecture, it improves over both of the two architectures. Note again that the final fusion weight employed for each feature is the product of the feature-level weight and the corresponding group-level weight. As a result, the final fusion weight combines the key information extracted from feature-based and group-based processing. Each group-level fusion weight can be reliably learned as in the FG-GFA architecture, as a result, the final fusion weight can more reliably reflect the importance of the corresponding feature with respect to the learning task at hand, and serves an effective gating mechanism for the feature. For example, as we have observed in our experimental study, the feature-level fusion weight for a noisy or corrupted sensory input may not fully reflect the degraded importance of that feature as in the case of the baseline architecture. The more reliable group-level fusion weight, however, can block this feature, i.e. by making the final feature weight (product of the feature-level and group-level weights) small. This property mitigates the issue of inconsistency of fusion weights of the baseline architecture.

On the other hand, compared with the FG-GFA architecture, 2S-GFA further leverages the information revealed by the feature-level fusion weights. Therefore, each sensory input can be gated at a finer granularity based on the feature-level fusion weight. As such, it may be expected that the 2S-GFA represents an optimized middle ground between the baseline netgated architecture and the coarser-grained FG-GPA architecture and that it can learn the structure of underlying data more effectively, as we will demonstrate experimentally.

## 5 EXPERIMENTAL SETTINGS

To validate the proposed FG-GFA and 2S-GFA architectures and compare them with the conventional non-gated and the baseline netgated architectures, we consider two applications: driving model prediction and human activity recognition on smartphones.

### 5.1 DATASETS, SETUPS FOR DATASETS, AND TOOL FLOW

**Dataset for driving model prediction.** We consider three driving modes: idle, eco, and normal. The idle mode corresponds to the situation in which the vehicle's engine is on but the vehicle is not in motion. The vehicle is in the eco mode when the car is being driven for fuel efficiency. All other situations are labeled as the normal mode. The target is to predict the vehicle's driving mode at the next time period given all the sensory inputs that have been available. We treat this application as a time-series prediction problem.

We have driven a 2014 Nissan Sentra for two months between two GPS coordinates to collect this driving dataset. The RPM and speed data are extracted from the on-board diagnostics (OBD2) in the vehicle. We use the y-axis reading of a gyroscope (GYRO_Y) for measuring lateral acceleration, and the difference between the GPS headings in time (D_HEADING) for the steering angle of a vehicle. All sensor data used in the driving data set are collected from Freematics ONE+ dongle(Huang, 2017). Due to the different extraction times periods of sensors in the Freematics ONE+ dongle, linear interpolation in time is used for data pre-processing. In total five types of sensory data are used for training the neural network: RPM, SPEED, acceleration and deceleration(D_SPEED), y axis of the gyroscope(GYRO_Y), and difference of GPS headings in time (D_HEADING). These five features are sampled every 250ms. To predict the driving mode of the vehicle 5 seconds in the future, we use ten seconds of data (40 points) from each feature, normalize the collected data feature-wise, and concatenate them to form a feature vector window. The feature vector window slides every 250ms as the prediction proceeds in time. For the proposed FP-GFA and 2S-GFA architectures, RPM, SPEED, and D_SPEED are included in the first group while the second group is composed of GYRO_Y, and D_HEADING. 5,845 examples are used for training, and 2,891 samples are used for testing We train different neural networks using 50,000 iterations.

**Dataset for human activity recognition.** We further consider the public-domain human activity recognition dataset(Anguita et al., 2013) which has six input features: three axes of an accelerometer (ACC_X, ACC_Y, ACC_Z), and three axes of a gyroscope (GYRO_X, GYRO_Y, GYRO_Z), and six activity classes: WALKING, WALKING_UPSTAIRS, WALKING_DOWNSTAIRS, SITTING, STANDING, and LAYING. These six features are sampled every 200ms. The same sliding window scheme is used to define input feature vectors. For the proposed group-level and two-stage gated architectures, the six features are split into two different groups. The first feature group has ACC_X and ACC_Y while ACC_Z. GYRO_X, GYRO_Y, and GYRO_Z are included in the second group. The training and test sets have 2,410 and 942 examples, respectively. 100,000 iterations are used for training various neural networks.

**Adopted tool flow.** The adopted simulation environment is based on Tensorflow 1.10.1(Abadi et al., 2015), a open source deep neural network library in Python.

### 5.2 CONFIGURATIONS OF FOUR COMPARED NEURAL NETWORKS

We compare the conventional CNN architecture, the baseline netgated architecture Patel et al. (2017), the proposed feature-group gated fusion architecture (FG-GFA), and the proposed two-stage gated fusion architecture (2S-GFA) by creating four corresponding neural network models. For fair comparison, we match the number of neurons and layers used in the processing common to all four networks as much as possible. Nevertheless, it shall be noted that compared with the CNN model, the netgated architecture has additional neurons for extracting feature-level fusion weights, FG-GFA employs additional neurons for extracting group-level fusion weights, and 2S-GFA employs both the feature and group level fusion weights. The configurations of the four neural networks are detailed in the Appendix.

## 5.3 SENSOR NOISE AND FAILURES

To more comprehensively evaluate the performance of different architectures, we employ the original data of the two adopted datasets, which are called "clean", and also create variations of the datasets by introducing sensory noise and failures for both the training and testing subsets of the data.

We mimic the degradation of sensory inputs by adding random Gaussian noise to all components of the feature vector

$$v_n = v_{ori}(1 + \gamma \varepsilon), \qquad (1)$$

where $v_{ori}$ and $v_n$ are the original and noisy component value, respectively, $\varepsilon$ is a random variable following the normal distribution $\sim N(0, 1)$, and $\gamma$ controls the percentage of the added noise and is set at different levels: 5%, 10%, and 20%.

In addition, we modify the original datasets by mimicking occurrence of more catastrophic sensor failures during both training and testing. Here, at each time stamp, one feature is selected at random and the corresponding value of that feature is wiped out to zero in the feature vector.

# 6 EXPERIMENTAL RESULTS

We evaluate the performance of different architectures using our driving mode prediction dataset and the public-domain human activity recognition dataset(Anguita et al., 2013) based on the settings described in the previous section.

## 6.1 RESULTS ON DRIVING MODE PREDICTION

**Predication accuracy with clean data.** Fig. 4 shows that the proposed two-stage architecture has the best prediction accuracy and the group-level FG-GFA architecture produces the second best performance when the data has no additional noise or failures. The two proposed architectures significantly improve over the conventional (non-gated) CNN architecture and also lead to noticeable improvements over the baseline netgated architecture.

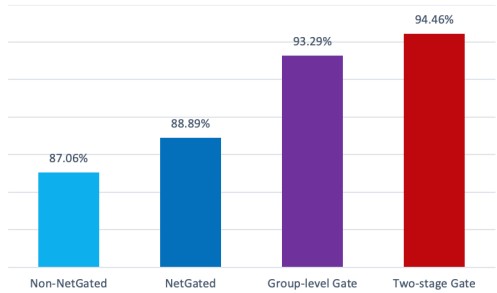

Figure 4: Results of driving mode prediction under clean data.

It is interesting to observe that while not producing the best performance for the testing set, the baseline netgated architecture has a loss less than those of the group-level and two-stage architectures for the training set as in Table 1, suggesting possible occurrence of over-fitting in the netgated architecture as discussed previously.

**Predication accuracy with noise or sensory failures.** To verify the robustness of the networks, we test four neural network models when different levels of Gaussian noises are introduced to the training and test data set. In Table. 2, the proposed two architectures produce robust performances with the two stage architecture being the best under all cases.

In Table. 3, we compare four different models under the introduction of random sensor failures. Since random failures are more catastrophic compared to Gaussian noise, the overall performances of all models drop in this case. Nevertheless, the proposed two models show the best performances and the two-stage architecture outperforms the baseline netgated architecture by nearly 3%.

Table 1: Training Losses of different models for driving model prediction under clean data.

| Network | Training Loss |
|---|---|
| Non-NetGated | 0.140 |
| NetGated | 0.145 |
| Group-level | 0.183 |
| Two-stage | 0.204 |

Table 2: Results of driving mode prediction with Gaussian noise.

| Noise level | Non-NetGated | NetGated | Group-level Gate | Two-stage |
|---|---|---|---|---|
| 5% | 87.54% | 88.80% | 91.48% | 93.28% |
| 10% | 83.46% | 86.23% | 90.21% | 92.01% |
| 20% | 83.20% | 86.05% | 90.19% | 91.32% |

Table 3: Results of driving mode prediction with sensor failures.

| Network | Prediction Accuracy |
|---|---|
| Non-NetGated | 85.47% |
| NetGated | 86.44% |
| Group-level | 88.34% |
| Two-stage | 89.20% |

**Analysis of performance improvements of the proposed architectures.** We provide additional insights on the possible causes for the observed performance improvements made by the proposed two-stage architecture. In our setup, the three essential features for driving mode prediction are RPM, speed, and D_SPEED, which are included in group 1. Table. 4 shows the feature and group level fusion weights of the two-stage architecture based on the clean data. We add 20% Gaussian noise to RPM in the group1 and report the updated fusion weights in Table.5. It can be seen that the feature-level fusion weight of RPM drops rather noticeably, possibly reflecting the degraded quality of this sensory input.

Table 4: Fusion weights of the two-stage architecture under clean driving mode data.

| | RPM | SPEED | D_SPEED | GYRO_Y | D_HEADING |
|---|---|---|---|---|---|
| Feature-level Fusion Weight | 0.26 | 0.18 | 0.16 | 0.21 | 0.19 |
| Group-level Fusion Weight | | 0.45 | | | 0.55 |

In a different experiment, we only add 20% noise to D_heading, which is a feature in the group2. As shown in Table.6, the feature-level fusion weight of D_heading and the group-level fusion weight of the second group both drop in comparison to the case of the clean data. It is expected that the reduced weights will reduce the contribution of D_heading to the final prediction decision.

## 6.2 HUMAN ACTIVITY RECOGNITION DATA SET

We adopt a similar approach to demonstrate the performances of various models on the human activity recognition data set.

**Predication accuracy with clean data.** Fig. 5 summarizes the performances of the four models based on the clean data. Again the two-stage 2S-GFA architecture produces the best performance and the proposed group-level architecture FG-GFA architecture produces the second best performance among the four models.

Table 5: Fusion weights of two-stage architecture under with 20% noise in RPM in the driving mode data.

|  | RPM | SPEED | D_SPEED | GYRO_Y | D_HEADING |
|---|---|---|---|---|---|
| Feature-level Fusion Weight | 0.15 | 0.22 | 0.18 | 0.27 | 0.18 |
| Group-level Fusion Weight |  | 0.57 |  |  | 0.43 |

Table 6: Fusion Weight Analysis with 20% of Noise in D_HEADING using two-stage architecture.

|  | RPM | SPEED | D_SPEED | GYRO_Y | D_HEADING |
|---|---|---|---|---|---|
| Feature-level Fusion Weight | 0.29 | 0.30 | 0.1 | 0.16 | 0.14 |
| Group-level Fusion Weight |  | 0.77 |  |  | 0.23 |

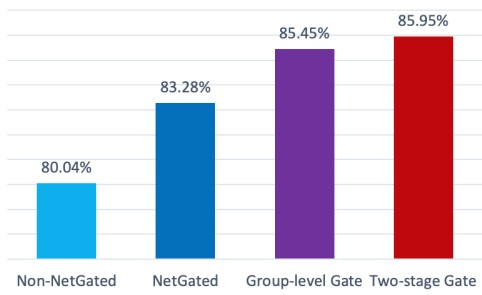

Figure 5: Results on human activity recognition under the clean data.

**Predication accuracy with noise or sensory failures.** Table. 8 shows with increasing Gaussian noise, the prediction accuracy of all models drops. However, the robustness and improvements of the two proposed architectures over the other two models are clearly observable. Table. 7 summarizes the results when sensor failures are introduced. In this case, the accuracy of the Non-netgated network model (conventional CNN) drops by 10%. Still, the two proposed architectures demonstrate improved robustness over the conventional CNN and the baseline netgated architectures. Specifically, for this more challenging test case, the two-stage gated architecture outperforms the non-netgated model by 5% and netgated model by 3%.

Table 7: Results on human activity recognition under sensor failures.

| Network | Prediction Accuracy |
|---|---|
| Non-NetGated | 70.16% |
| NetGated | 75.37% |
| Group-level Gate | 77.17% |
| Two-stage Gate | 78.02% |

Table 8: Results on human activity recognition under Gaussian noise.

| Noise level | Non-NetGated | NetGated | Group-level Gate | Two-stage |
|---|---|---|---|---|
| 5% | 78.45% | 82.53% | 84.18% | 84.92% |
| 10% | 77.49% | 82.50% | 83.43% | 83.90% |
| 20% | 76.61% | 80.40% | 82.05% | 82.91% |

## 7 CONCLUSION

This paper proposes two optimized gated deep learning architectures based on CNNs for sensor fusion: a coarser-grained gated architecture with (feature) group-level fusion weights and a two-stage architecture with a combination of feature-level and group-level fusion weights. It has been shown that the proposed architectures outperform the conventional CNN architecture and the existing Net-Gated architecture under various settings. Especially, the proposed architectures demonstrate larger improvements in the presence of random sensor noise and failures. Our future work will extend the proposed architectures for more complex sensor applications which may include additional sensing modalities such as cameras and Lidars.

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

# A APPENDICES

## A.1 NEURAL NETWORK ARCHITECTURES USED FOR DRIVING MODEL PREDICTION

### A.1.1 NON-NETGATED ARCHITECTURE

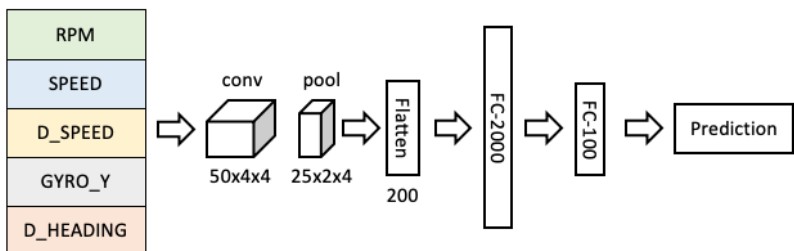

Figure 6: Non-NetGated CNN for driving model prediction.

### A.1.2 NETGATED ARCHITECTURE

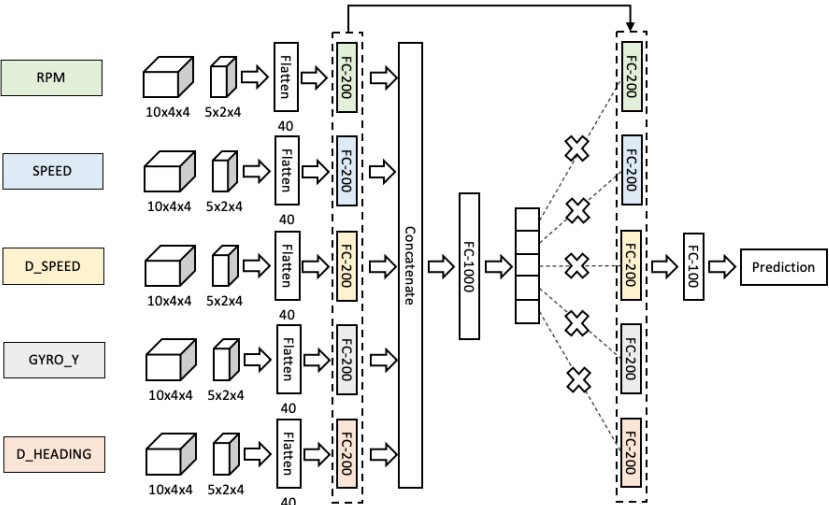

Figure 7: NetGated CNN for driving model prediction.

### A.1.3   THE PROPOSED FEATURE-GROUP GATED FUSION ARCHITECTURE

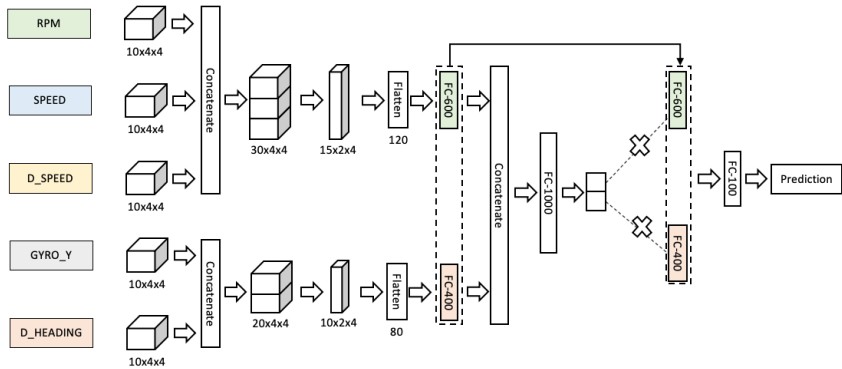

Figure 8: The proposed feature-group gated CNN for driving model prediction.

### A.1.4   THE PROPOSED TWO-STAGE GATED FUSION ARCHITECTURE

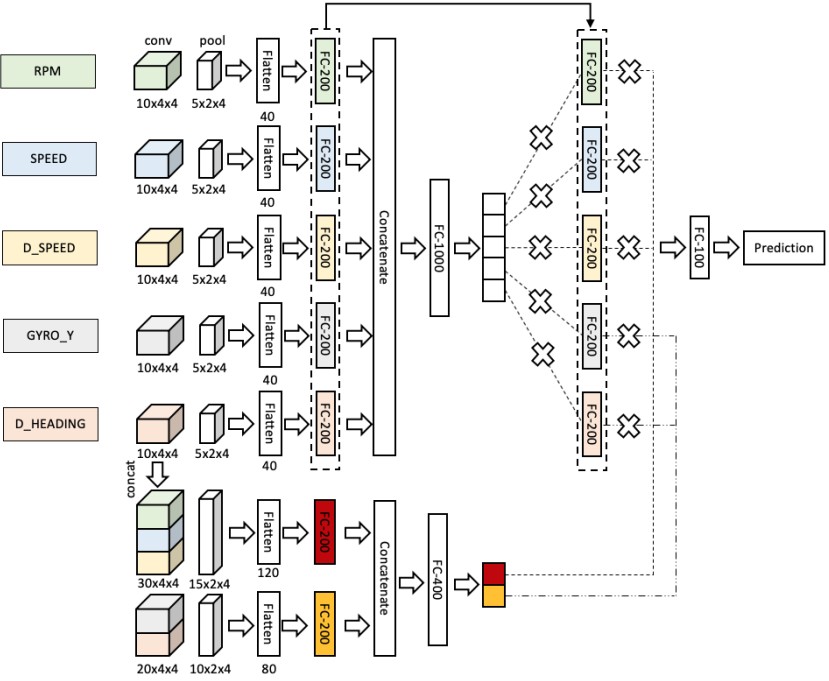

Figure 9: The proposed two-stage gated CNN for driving model prediction.

## A.2 NEURAL NETWORK ARCHITECTURES USED FOR HUMAN ACTIVITY RECOGNITION

### A.2.1 NETGATED ARCHITECTURE

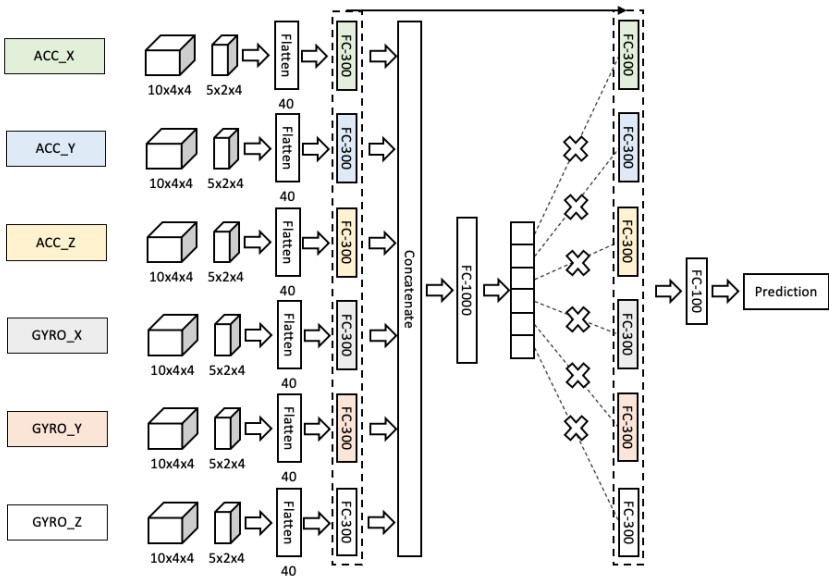

Figure 10: Netgated CNN for human activity recognition.

### A.2.2 THE PROPOSED FEATURE-GROUP GATED FUSION ARCHITECTURE

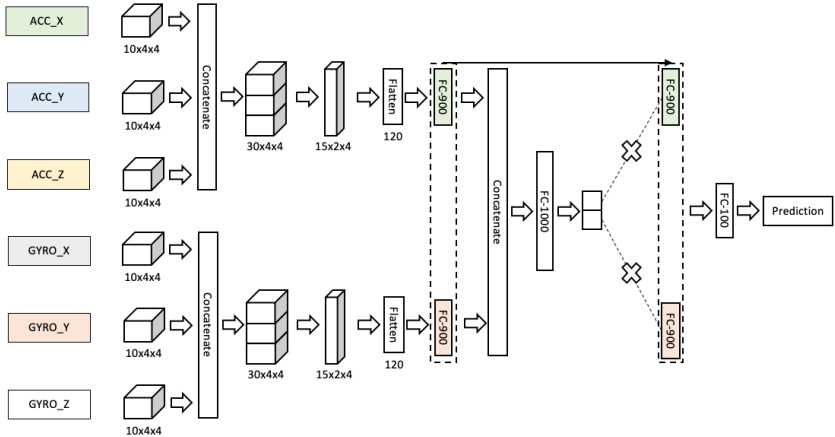

Figure 11: The proposed feature-group gated CNN for human activity recognition.

### A.2.3 The Proposed Two-stage Gated Fusion Architecture

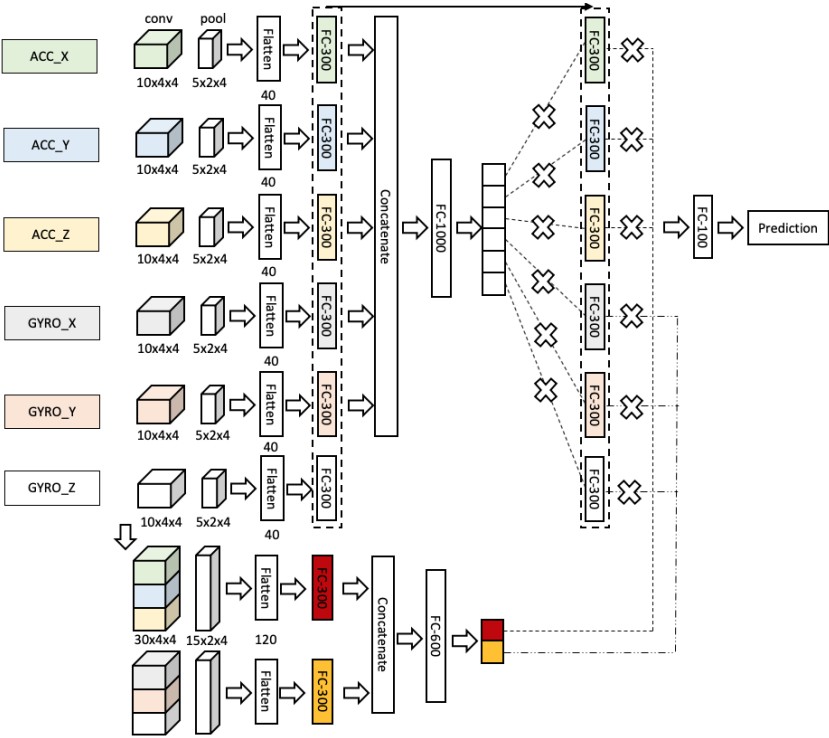

Figure 12: The proposed two-stage gated CNN for human activity recognition.

