# OpenReview forum: "Optimized Gated Deep Learning Architectures for Sensor Fusion"
_ICLR.cc/2019/Conference_

### Official Review · AnonReviewer2 · 2018-11-02
**#Summary**

**Rating:** 3
**Confidence:** 4

**Review:**

This paper proposes two gated deep learning architectures for sensor fusion. They are all based on the previous work
Naman Patel et al's modality fusion with CNNs for UGV autonomous driving in indoor environments (IROS). By having the grouped features, the author demonstrated improved performance, especially in the presence of random sensor noise and failures.

#Organization/Style:
The paper is well written, organized, and clear on most points. A few minor points:
1) The total length of the paper exceeds 8 pages. Some figures and tables should be adjusted to have it fit into 8 pages.
2) The literature review is limited.
3) There are clearly some misspellings. For example, the "netgated" is often written as "negated".

#Technical Accuracy:
The two architecture that the author proposes all based on the grouped features, which to my point of view, is a very important and necessary part of the new model. However, the author failed to rigorously prove or clearly demonstrated that why this is effective to our new model.  Moreover, how to make groups or how many groups are needed are not clearly specified. The experiments used only two completely different datasets, none of them are related to the previous sensor fusion method they are trying to compete. I'm afraid this method cannot generalize to a common case.

In addition, if we look at Table 4 and Table 5, we can find the first Group-level Fusion Weight actually increases, which seems contradictory to the result shown in Table 6.

#Adequacy of Citations:
Poor coverage of literature in sensor fusion. There are less than 10 references are related to sensor fusion.

Overall, it is not an ICLR standard paper.

---

### Official Review · AnonReviewer1 · 2018-11-03
**2 new architectures for multisensor fusion, with improved results on standard settings and also with noisy/missing modalities**

**Rating:** 4
**Confidence:** 4

**Review:**

Overview and contributions: The authors improve upon several limitations of the baseline negated architecture by proposing 1) a coarser-grained gated fusion architecture and 2) a two-stage gated fusion architecture. The authors show improvements in driving mode prediction and human activity recognition in settings where all modalities are observed as well as settings where there are noisy or missing modalities.

Strengths:
1. The model seems interesting and tackles the difficult problem of multisensor fusion under both normal and noisy settings.
2. Good results obtained on standard benchmarks with improvements in settings where all modalities are observed as well as settings where there are noisy or missing modalities.

Weaknesses:
1. I am worried about the novelty of the proposed approach. The main idea for the fusion-group gated fusion architecture is to perform additional early fusion of sensory inputs within each group which reduces the number of group-level fusion weights and therefore the number of parameters to tune. The two-stage gated fusion architecture simply combines the baseline model and the proposed fusion-group model. Both these ideas seem relatively incremental.
2. Doesn't the final two-stage gated fusion architecture further increase the number of parameters as compared to the baseline model? I believe there are several additional FC-NN blocks in Figure 3 and more attention gating weights. I find this counterintuitive since section 2.2 motivated "Potential Over-fitting" as one drawback of the baseline Netgated architecture. How does the increase in parameters for the final model affect the running time and convergence?

Questions to authors:
1. I don't understand Tables 4,5,6. Why are the results for Group-level Fusion Weight in the middle of several columns? Which features are being used in which groups? Please make this clear using vertical separators.
2. For the proposed two-stage gated fusion architecture, do the 2 branches learn different things (i.e focus on different portions of the multimodal inputs)? I would have liked to see more visualizations and analysis instead of just qualitative results.

Presentation improvements, typos, edits, style, missing references:
1. General poor presentation of experimental results. Tables are not clear and bar graphs are not professionally drawn. The paper extends to 9 pages when a lot of space could be saved by making the presentation of experimental results more compact. I believe the guidelines mention that more pages can be used if there are extensive results, but I don't think the experimental results warrant the extra page.

---

### Official Review · AnonReviewer3 · 2018-11-04
**Interesting Topic but Lacks Insight, Novely, and Experimental Rigor**

**Rating:** 4
**Confidence:** 5

**Review:**


This paper tackles the problem of sensor fusion, where multiple (possibly differing) sensor modalities are available and neural network architectures are used to combine information from them to perform prediction tasks. The paper proposed modifications to a gated fusion network specifically: 1) Grouping sets of sensors and concatenating them before further processing, and 2) Performing multi-level fusion where early sensor data representations are concatenated to produce weightings additional to the those obtained from features concatenated at a later stage. Experimental results show that these architectures achieve performance gains from 2-6%, especially when sensors are noisy or missing.

Strengths

 + The architectures encourage fusion at multiple levels (especially the second one), which is a concept that has been successful across the deep learning literature

 + The paper looks at an interesting topic, especially related to looking at the effects of noise and missing sensors on the gating mechanisms.

 + The results show some positive performance gains, although see caveats below.

Weaknesses

 - The related work paragraph is extremely sparse. Fusion is an enormous field (see survey cited in this paper as well [1]), and I find the small choice of fusion results with a YouBot to be strange. A strong set of related work is necessary, focusing on those that are similar to the work. As an example spatiotemporal fusion (slow fusion [2]) bears some resemblence to this work but there are many others (e.g. [3,4] as a few examples).

   [1] Ramachandram, Dhanesh, and Graham W. Taylor. "Deep multimodal learning: A survey on recent advances and trends." IEEE Signal Processing Magazine 34.6 (2017): 96-108.
 	   Ramach
   [2] Karpathy, Andrej, et al. "Large-scale video classification with convolutional neural networks." Proceedings of the IEEE conference on Computer Vision and Pattern Recognition. 2014
   [3] Mees, Oier, Andreas Eitel, and Wolfram Burgard. "Choosing smartly: Adaptive multimodal fusion for object detection in changing environments." Intelligent Robots and Systems (IROS), 2016 IEEE/RSJ International Conference on. IEEE, 2016.
   [4] Kim, J., Koh, J., Kim, Y., Choi, J., Hwang, Y., & Choi, J. W. (2018). Robust Deep Multi-modal Learning Based on Gated Information Fusion Network. arXiv preprint arXiv:1807.06233.

 - The paper claims to provide a "deep understanding of the relationships between sensory inputs, fusion weights, network architecture, and resulting performance". I don't think it really achieves
    this with the small examples of weights for some simple situations.

 - It is very unclear whether the architectures have more or less parameters. At one point it is stated that the original architecture overfits and the new architecture has less parameters (Sec 2.2 and 3). But then it is stated for fairness the number of neurons is equalized (5.2), and later in that section that the new architectures have additional neurons. Which of these is accurate?

 - Related to the previous point, and possibly the biggest weakness, the experimental methodology makes it hard to tell if performance is actually improved. For example, it is not clear to me that the performance gains are not just a result of less overfitting (for whatever reason) of baselines and that the fixed number of epochs therefore results in stopping at a better performance. Please show training and validation curves so that we can see whether the epochs chosen for the baselines are not just chosen after overfitting (in which case early stopping will improve the performance). As another example, there are no variances shown in the bar graphs.

 - The examples with noise and failures are limited. For example, it is also not clear why an increase of noise in the RPM feature (Table 5) actually increases the weight of that group in the two-stage architecture. What does that mean? In general there isn't any principled method proposed for analyzing these situations.

Some minor comments/clarifications:
  - What is the difference between these gated networks and attentional mechanisms, e.g. alpha attention (see "Attention is all you need" paper)?
  - What is a principled method to decide on the groupings?
  - There are several typos throughout the paper
    * "in the presence of snesor" => "in the presence of sensor"
    * Throughout the paper: "Predication" => "Prediction"
    * "Likelihood of stucking the training"
  - Tensorflow is not a simulation environment

Overall, the paper proposes architectural changes to an existing method for fusion, and while positive results are demonstrated there are several issues in the experimental methodology that make it unclear where the benefits come from. Further, the paper lacks novelty as multi-level fusion has been explored significantly and the changes are rather minor. There is no principled method or concepts that drive the architectural changes, and while the authors claim a deeper investigation into the networks' effectiveness under noise and failures the actual analysis is too shallow.

---

### Meta-Review · Area_Chair1 · 2018-12-12
**good work, but not ripe enough**

**Confidence:** 5
**Recommendation:** Reject

**Metareview:**

The paper builds up on the gated fusion network architectures, and adapt those approaches to reach improved results.  In that it is incrementally worthwhile.

All the same, all reviewers agree that the work is not yet up to par.  In particular, the paper is only incremental, and the novelty of it is not clear.  It does not relate well to existing work in this field, and the results are not rigorously evaluated; thus its merit is unclear experimentally.